# Bi-directional Deformation for Parameterization of Neural Implicit Surfaces

## Abstract

The growing capabilities of neural rendering have increased the demand for new techniques that enable the intuitive editing of 3D objects, particularly when they are represented as neural implicit surfaces. In this paper, we present a novel neural algorithm to parameterize neural implicit surfaces to simple parametric domains, such as spheres, cubes or polycubes, where 3D radiance field can be represented as a 2D field, thereby facilitating visualization and various editing tasks. Technically, our method computes a bi-directional deformation between 3D objects and their chosen parametric domains, eliminating the need for any prior information. We adopt a forward mapping of points on the zero level set of the 3D object to a parametric domain, followed by a backward mapping through inverse deformation. To ensure the map is bijective, we employ a cycle loss while optimizing the smoothness of both deformations. Additionally, we leverage a Laplacian regularizer to effectively control angle distortion and offer the flexibility to choose from a range of parametric domains for managing area distortion. Designed for compatibility, our framework integrates seamlessly with existing neural rendering pipelines, taking multi-view images as input to reconstruct 3D geometry and compute the corresponding texture map. We also introduce a simple yet effective technique for intrinsic radiance decomposition, facilitating both view-independent material editing and view-dependent shading editing. Our method allows for the immediate rendering of edited textures through volume rendering, without the need for network re-training. Moreover, our approach supports the co-parameterization of multiple objects and enables texture transfer between them. We demonstrate the effectiveness of our method on images of human heads and man-made objects. We will make the source code publicly available.

## 1 Introduction

Neural radiance fields (NeRF)(Mildenhall et al., 2020) have garnered remarkable success in both the computer vision and computer graphics communities, redefining benchmarks for high-quality renderings and novel view synthesis. Building upon NeRF, a variety of methods leveraging implicit neural representations (Oechsle et al., 2021; Wang et al., 2021; Yariv et al., 2021) have emerged, delivering high-fidelity 3D reconstructions. The demand for intuitive 3D object editing techniques has increased with the growing capabilities of neural rendering, especially for neural implicit surfaces. Extensive explorations have been made in editing within the Neural Rendering framework to achieve diverse visual effects. We can implicitly alter the shape and color of 3D objects by manipulating the latent codes of the network(Liu et al., 2021; Yenamandra et al., 2021; Xu et al., 2023). In the work of (Tojo & Umetani, 2022; Kuang et al., 2023), palette-based methods have been employed to facilitate transformations in the color and style of rendered scenes. However, these methods suffer from limited controllability and disallow local pixel editing. Seal3D (Wang et al., 2023) proposed a teacher-student training strategy to edit the interactive scene at the pixel level. Yet, this method is tricky to handle shading, and each subsequent edit required a new training iteration. By employing neural networks to formulate a differentiable surface parameterization, it can be facilitated to edit textures or shapes within the neural rendering pipeline (Xiang et al., 2021; Ma et al., 2022). Existing neural parameterization techniques still face significant challenges, such as the need for appropriate distortion constraints to ensure accurate detail reconstruction or reducing the reliance on prior infor-

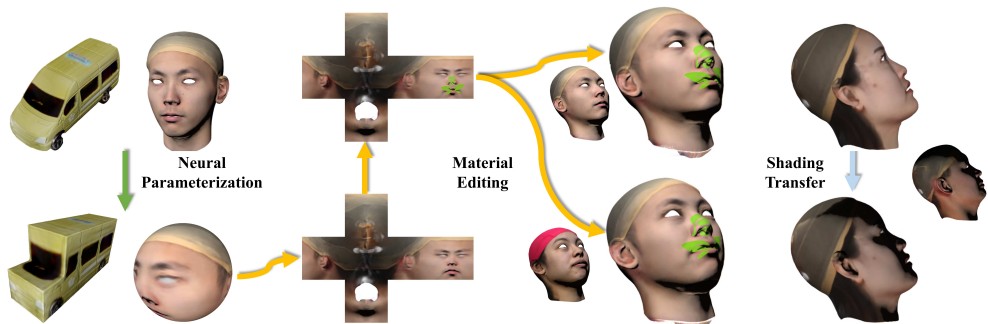

Figure 1: Our method takes multi-view images as input to reconstruct 3D geometry as neural implicit surfaces, while representing appearance through a mapping between the 3D surface and a user-selected parametric domain. Such a representation allows volume rendering of modified textures without the need of network retraining, facilitating both view-dependent shading editing and view-independent material editing.

mation from tracked mesh and UV mapping. The final quality of edited visual effects is significantly impacted by these issues.

This paper aims at developing an intuitive and easy-to-use tool for appearance editing in neural implicit surfaces. Towards this goal, we propose a novel method to parameterize neural implicit surfaces to parametric domains, such as spheres, cubes or polycubes, where the radiance field is represented as a texture map. Technically, our parameterization algorithm utilizes a neural network to learn a bi-directional deformation between 3D objects and the chosen parametric domains. This involves a forward deformation that maps points from the zero level set of the neural implicit surface to the parametric domain, followed by an inverse deformation, mapping points backward. Notably, we do not require any explicit prior information in learning both deformations. We employ a cycle loss to ensure the smoothness of the bi-directional deformation, and a Laplacian regularization to effectively control angle distortion. Our method also supports the flexible selection of parametric domains for controlling area distortion. With the parametrization, 3D radiance field becomes essentially a 2D field, facilitating visualization and various editing tasks. we further decompose the 2D radiance field into two components: view-independent material and view-dependent shading, streamlining both texture and shading editing. Our neural parameterization algorithm is fully compatible with existing neural rendering pipelines, allowing 3D reconstruction from multi-view images as input and the creation of texture maps simultaneously. It allows for the immediate rendering of edited textures through volume rendering, without the need for network re-training. Moreover, it also supports co-parameterization of multiple objects of similar geometry and enable texture transfer between them. We validate the effectiveness of our method on images of human heads and man-made objects.

Our contributions are summarized as follows:

- We present a neural parameterization framework that computes bi-directional deformation between 3D objects and their parametric domains, eliminating the need for any prior information from tracked mesh or UV mapping. Our approach utilizes Laplacian loss to minimize angle distortion and provides a choice of parametric domains, such as spheres, cubes, or polycubes, to control area distortion.

- We introduce a simple yet effective technique for intrinsic radiance decomposition, facilitating both view-independent texture editing and view-dependent shading editing.

- Our framework is fully compatible with existing neural rendering architectures, accepting multi-view images as input to reconstruct 3D geometry and generate UV maps as output. Additionally, it enables the direct rendering of modified textures using volume rendering pipeline.

- Our approach supports co-parameterization of multiple objects and allows for texture transfer between different objects.

## 2 RELATED WORK

**Parameterization.** Surface parameterization (Sheffer et al., 2007; Floater & Hormann, 2005) aims at computing a bijective mapping between the 3D surface and a suitable parametric domain, usually a 2D region or a simple 3D object, such as spheres (Gotsman et al., 2003) and polycubes (García et al., 2013). Serving as an important computational tool in computer graphics and digital geometry processing (Sheffer et al., 2007), surface parameterization facilitates various applications, including texture editing (Fang & Hart, 2004), surface painting (Sun et al., 2013), details transfer (Biermann et al., 2002), remeshing (Praun & Hoppe, 2003), among others. Neural networks are now commonly used in digital geometry processing, resulting in the creation of parameterization algorithms based on deep learning techniques (Groueix et al., 2018; Williams et al., 2019; Bednarik et al., 2020; Guo et al., 2022). Unlike classical methods that yield global and seamless parametrization, most neural parameterization methods are computing local parametrization (Groueix et al., 2018; Zhang et al., 2023; 2022). These methods typically partition the 3D model into multiple patches and parameterize each to a 2D region, possibly followed by a post-processing step to stitch the patches together. Although these methods are efficient and can handle surfaces of arbitrary geometry and topology, they frequently encounter problems including non-bijectivity, a lack of smoothness, the presence of seams and overlaps, and large distortions. There are also works on neural parametric surfaces, defined in rectangular domains (Low & Lee, 2022) and $n$-sided patches (Yang et al., 2023). They can model complex surface geometries with high precision; however, such representations are not applicable to neural implicit surfaces and thereby incompatible with neural rendering.

**Neural Implicit Functions.** Recent years have seen a surge in the successful application of neural implicit functions within the realm of 3D deep learning. Compared to explicit representations such as point clouds (Qi et al., 2017), voxels (Qi et al., 2016), and meshes (Wang et al., 2018), neural implicit representations offer unique advantages including flexibility, continuity, and robustness (Mescheder et al., 2019; Saito et al., 2019; Park et al., 2019). When integrated within the neural rendering pipeline, neural implicit surfaces exhibit exceptional quality in 3D reconstructions from multi-view images (Li & Zhang, 2021; Wang et al., 2022a;b; Xu et al., 2023; Müller et al., 2022; Li et al., 2023; Rosu & Behnke, 2023). However, as geometries and radiance fields are encoded as network parameters, editing them becomes complex and non-intuitive, compared to explicit representations. This paper aims at tackling this challenge by explicitly representing radiance fields as texture mapping that encompasses both view-dependent and view-independent components.

**NeRF Editing.** Numerous studies have focused on editing neural radiance fields, including relighting (Srinivasan et al., 2021; Li et al., 2022), composition (Pérez et al., 2023; Yang et al., 2021), content generation (Niemeyer & Geiger, 2021), and shape editing (Yuan et al., 2023; Lin et al., 2022). Editing tools operating at the scene- or object-level, such as (Yang et al., 2021; Yenamandra et al., 2021; Huang et al., 2022), utilize latent codes to modify and stylize the appearance of an object or entire scenes. Conversely, pixel-level editing tools, such as Seal3D (Wang et al., 2023) and NeuMesh (Yang et al., 2022), utilize training-based approaches to edit fine-grained details. However, they necessitate retraining for each editing, lacking efficiency and robustness. Parameterization-based methods, such as NeuTex (Xiang et al., 2021), ISO-UVField (Sagnik Das & Samaras, 2022) and NeP (Ma et al., 2022), integrate the learning of UV mapping into the neural rendering framework, providing an intuitive means for editing within 2D domains. However, these methods often suffer from large distortions in the parametrization. Furthermore, they require 3D prior information from tracked meshes and UV mapping - conditions that may not be readily met in practical, real-world scenarios. IntrinsicNeRF (Ye et al., 2023) enhances standard neural radiance fields by generating additional outputs, including reflectance, shading, and a residual term. Incorporating a semantic branch, it facilitates real-time scene editing. However, it often results in aggregated colors, akin to palette-based methods (Tojo & Umetani, 2022; Kuang et al., 2023), hence cannot offer fine-grained editing results at the pixel level. Our approach decomposes the radiance field into view-dependent shadings and view-independent materials. Both components are represented as textures, enabling intuitive material modifications while preserving consistent shading.

## 3 METHOD

The input of our method is a collection of RGB images, denoted $\mathcal{I} = \{\mathbf{I}_i\}$, representing either multiple objects of similar geometry or a single object, captured from different viewpoints. To

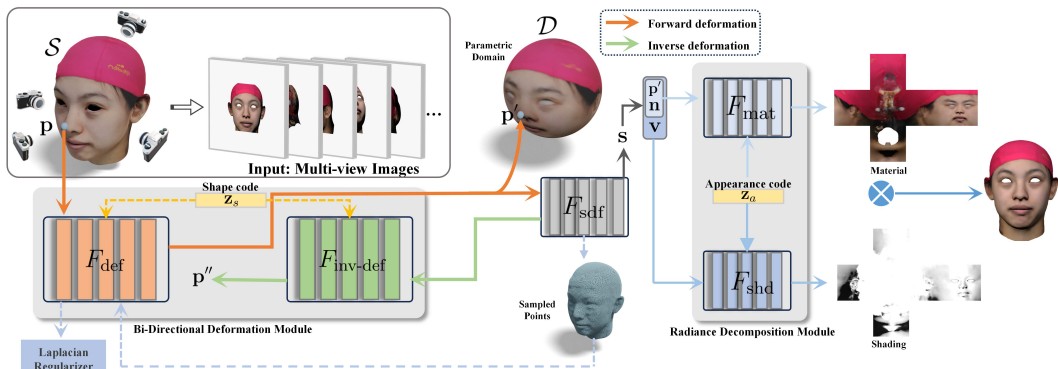

Figure 2: Algorithmic pipeline. Our network consists of three modules: bi-directional deformation ($F_{\text{def}}$ and $F_{\text{inv-def}}$), geometry reconstruction ($F_{\text{sdf}}$) and radiance decomposition ($F_{\text{mat}}$ and $F_{\text{shd}}$).

differentiate among objects, we assign a latent shape code $\mathbf{z}_s$ and an appearance code $\mathbf{z}_a$ to each object. This facilitates the co-parameterization of multiple objects and enables transfer material and shading among them. As illustrated in Figure 2, our method reconstructs 3D geometry as a neural implicit surface and encodes the 3D radiance field in a simple parametric domain, such as spheres and polycubes. For each input image, we randomly select a set of pixels. Then, for each pixel, we shoot a ray originating from the location of camera and traversing through the pixel. If the ray intersects with the surface, at a point, denoted as $p$, we compute its corresponding position in the parametric domain $\mathcal{D}$, denoted as $\mathbf{p}'$, via a bi-directional deformation (Sec. 3.1): the forward deformation (depicted in orange) generates a displacement vector so that the updated position $\mathbf{p}'$ is in the parametric domain; subsequently, the inverse deformation (depicted in green) maps $\mathbf{p}'$ back to a point $\mathbf{p}''$, which is anticipated to be in close proximity to the surface. We incorporate a cycle loss to penalize occurrences where $\mathbf{p}''$ deviates from $\mathbf{p}$. In addition, we employ a Laplacian loss (Sec. 3.2) to effectively reduce the angle distortion. Furthermore, we decompose the radiance into a view-independent material field and a view-dependent shading field (Sec. 3.3), each of which can be edited independently, thus augmenting the editing capability of our framework.

### 3.1 BI-DIRECTIONAL DEFORMATION

The bi-directional deformation module serves as the pivotal component within our neural parameterization framework. As illustrated in Figure 2, this module consists of two sub-networks: $F_{\text{def}}$, which is responsible for the forward mapping from the original model to the parameter domain, and $F_{\text{inv-def}}$, which establishes the inverse mapping. Let $\mathcal{S}$ be the 3D surface and $\mathcal{D}$ be the parametric domain; both are represented by the zero level-set of neural signed distance fields. For an arbitrary point $\mathbf{p} \in \mathcal{S}$, the forward deformation $F_{\text{def}}(\mathbf{p}, \mathbf{z}_s)$ computes a displacement vector, mapping the point $\mathbf{p}$ to a corresponding point $\mathbf{p}' \in \mathcal{D}$ in the parametric domain as follows:

$$\mathbf{p}' = \mathbf{p} + F_{\text{def}}(\mathbf{p}, \mathbf{z}_s). \tag{1}$$

Conversely, we employ $F_{\text{inv-def}}$ to map points inversely from the parametric domain $\mathcal{D}$ back to the original shape $\mathcal{S}$:

$$\mathbf{p}'' = \mathbf{p}' + F_{\text{inv-def}}(\mathbf{p}', \mathbf{z}_s), \tag{2}$$

where the point $\mathbf{p}''$ is expected to be in close proximity to $p$. To encourage the smoothness and stability of the deformations, we impose an $L_2$ loss on the magnitude of the displacement vectors:

$$\mathcal{L}_{\text{smooth}} = \|F_{\text{def}}(\mathbf{p}, \mathbf{z}_s)\|_2 + \|F_{\text{inv-def}}(\mathbf{p}', \mathbf{z}_s)\|_2. \tag{3}$$

This loss helps minimize abrupt and irregular deformations, leading to more coherent and gradual changes between $\mathcal{S}$ and $\mathcal{D}$.

In training the networks $F_{\text{def}}$ and $F_{\text{inv-def}}$, our objective is to achieve a bijective mapping between the 3D surface $\mathcal{S}$ and the parametric domain $\mathcal{D}$. To achieve this goal, we adopt a cycle loss, which is similar to NeuTex (Xiang et al., 2021):

$$\mathcal{L}_{\text{cycle}} = \sum_{\mathbf{p} \in \mathcal{S}} \lambda(\mathbf{p}) \|\mathbf{p} - \mathbf{p}''\|_2, \tag{4}$$

where the weight $\lambda$ characterizes the importance of sample $\mathbf{p}$ to the loss. The reason that we do not consider all sample points equally is that a sample point may not be exactly on the zero level-set of the SDF. To tolerate such an inaccuracy, we define $\lambda(\mathbf{p}) = T(\mathbf{p})(1 - \exp(-\sigma(s)\delta(\mathbf{p})))$ as the color weight used in volume rendering (Mildenhall et al., 2020). Here, $T(\mathbf{p})$ is the transparency of sample $\mathbf{p}$ in the viewing direction, $\delta$ is the length of the sample interval along the ray, and $s$ is the signed distance value of the point $\mathbf{p}$. Clearly, when $\mathbf{p}$ is away from the surface, the coefficient $\lambda(\mathbf{p})$ has few effects on this loss. We refer the readers to (Mildenhall et al., 2020) for details about volume rendering and the computation of transparency $T$. By explicitly minimizing the distance between $\mathbf{p}$ and $\mathbf{p}''$, the cycle loss effectively prevents scenarios where two distinct points $\mathbf{p}_1, \mathbf{p}_2 \in \mathcal{S}$ map to an identical point $\mathbf{p}' \in \mathcal{D}$. In such cases, the inverse deformation $F_{\text{inv-def}}$ would be unable to map the single point $\mathbf{p}'$ back to the respective distinct points $\mathbf{p}''_1$ and $\mathbf{p}''_2$.

**Remark.** It is worth mentioning that while our framework involves two SDFs - one for the 3D surface $\mathcal{S}$ and the other for the parametric domain $\mathcal{D}$ - our network design only requires one of them. This is because one SDF can be derived from the other via either the forward or the backward deformation. To reduce the network complexity, we only adopt one geometry sub-network $F_{\text{sdf}}$, which is used to represent the parametric domain $\mathcal{D}$. Given a sphere, a cube, or a polycube, we pretrain the SDF sub-network to encode $\mathcal{D}$'s geometry. Leveraging the inherent smoothness of MLPs, the resulted SDF is smooth, aiding the computation of Laplacian.

### 3.2 LAPLACIAN REGULARIZATION

Mitigating angle distortion is crucial in ensuring the quality of parameterization. One approach to reduce angle distortion involves minimizing the ratio of the two singular values of the Jacobian matrix associated with the parameterization (Floater & Hormann, 2005).

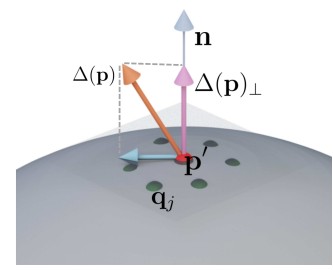

However, considering the representation of the surface $\mathcal{S}$, the parametric domain $\mathcal{D}$, and the parameterization $F_{\text{def}}$ as neural implicit functions in our approach, directly computing the Jacobian matrix is a non-trivial task. To circumvent this, we employ a Laplacian regularizer in our implementation to control angle distortion. Achieving the minimizer of the Laplacian of the map results in a harmonic map. For genus-0 surfaces, harmonic maps are also conformal, which preserves angles (Gu et al., 2004). Although for surfaces of higher genus, a harmonic map is not necessarily conformal, it still serves an effective tool for reducing angle distortion (Gu & Yau, 2003). Specifically, for a sample point $\mathbf{p} \in \mathcal{S}$ and its neighboring points $\mathbf{q}_j \in \mathcal{S}$, the forward deformation maps them to $\mathbf{p}' \in \mathcal{D}$ and $\mathbf{q}'_j \in \mathcal{D}$, respectively. We compute the Laplacian of the sample $\mathbf{p}$ as follows:

$$\Delta(\mathbf{p}) = \sum_{j=1}^{m} \omega_j (\mathbf{p}' - \mathbf{q}'_j). \tag{5}$$

Here, the weights $w_j$ are defined using the metric from the original surface $\mathcal{S}$ as:

$$\omega_j = \exp\left(-\frac{\|\mathbf{p} - \mathbf{q}_j\|_2}{l}\right), \tag{6}$$

where $l$ is the average distance from point $\mathbf{p}$ to its neighboring points. Subsequently, the weights are normalized $\omega_j = \omega_j / \sum_{k=1}^{m} \omega_k$. Notice that $\Delta(\mathbf{p}) \in \mathbb{R}^3$ is a 3D vector, which may not necessarily lie on the tangent plane. Therefore, we define the normal component of the Laplacian as:

$$\Delta(\mathbf{p})_\perp = \langle \Delta(\mathbf{p}), \mathbf{n} \rangle \mathbf{n}, \tag{7}$$

where $\mathbf{n}$ is the normal at $\mathbf{p}'$, and $\langle , \rangle$ represents the inner product. Summing the tangential components yields the Laplacian loss:

$$\mathcal{L}_{\text{Lap}} = \sum_{\mathbf{p} \in \mathcal{S}} \|\Delta(\mathbf{p}) - \Delta(\mathbf{p})_\perp\|_2. \tag{8}$$

In our implementation, we set $m = 6$ neighboring points for each sample $\mathbf{p}$.

### 3.3 APPEARANCE DECOMPOSITION

Radiance represents the directional emission of color, illustrating how the appearance of an object varies in response to changes in the observational viewpoint (Mildenhall et al., 2020). Drawing inspiration from previous works (Ma et al., 2022; Ye et al., 2023) and applying the assumptions of Lambertian and grayscale shading (Fan et al., 2018), we decompose the radiance field of the target surface into two components: a view-independent material field and a view-dependent shading field (see Figure 7(b)). This decomposition allows for independent editing of material and shading. More precisely, for a sample point $\mathbf{p} \in \mathcal{S}$, we employ two sub-networks, $F_{\text{mat}}$ and $F_{\text{shd}}$, to decompose the radiance $\mathbf{r} \in \mathbb{R}^3$ at point $\mathbf{p}$ as follows:

$$\mathbf{r}(\mathbf{p}) = F_{\text{mat}}(\mathbf{p}', \mathbf{n}, \mathbf{z}_a) \cdot \exp(F_{\text{shd}}(\mathbf{p}', \mathbf{n}, \mathbf{v}, \mathbf{z}_a)), \tag{9}$$

where $\mathbf{p}' \in \mathcal{D}$ is the image of $\mathbf{p}$ and $\mathbf{n}$ is the unit normal of $\mathbf{p}$ on the original surface. The material network $F_{\text{mat}}$ takes the mapped point $\mathbf{p}'$, the surface normal $\mathbf{n}$, and the appearance latent code $\mathbf{z}_a$ as input, generating the view-independent material field in response. The shading network $F_{\text{shd}}$, which is designed for computing the view-dependent shading field, takes an additional input: the viewpoint $\mathbf{v}$. To ensure the shading sub-network $F_{\text{shd}}$ concentrates on **local** shading features, we employ an $L_1$ loss to the output of $F_{\text{shd}}$, encouraging sparsity,

$$\mathcal{L}_{\text{shd}} = \|F_{\text{shd}}(\mathbf{p}', \mathbf{n}, \mathbf{v}, \mathbf{z}_a)\|_1. \tag{10}$$

To compute the color, we need the signed distance values for the samples $\mathbf{p}$. As mentioned above, our network adopts only one SDF network $F_{\text{sdf}}$, which represents the geometry of the parametric domain $\mathcal{D}$. We use the forward deformation $F_{\text{def}}$ to compute $s(\mathbf{p})$ as

$$s(\mathbf{p}) = F_{\text{sdf}}\left(\mathbf{p} + F_{\text{def}}(\mathbf{p}, \mathbf{z}_s)\right). \tag{11}$$

Then we employ the SDF-based volume rendering (Yariv et al., 2021) to integrate the radiances along the ray to compute color $\mathbf{c}_{\text{pred}} = \sum_i \lambda_i \mathbf{r}_i$, where $\mathbf{r}_i$ is the radiance of a sample point $\mathbf{p}_i$ and $\lambda_i$ is the same color weight as used in Equation (4).

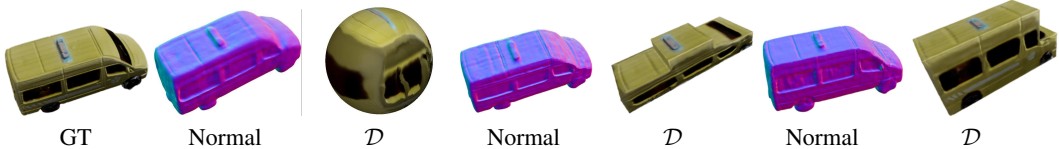

GT      Normal      $\mathcal{D}$      Normal      $\mathcal{D}$      Normal      $\mathcal{D}$

Figure 3: We offer the flexibility to choose from a range of parametric domains for managing area distortion. From left to right: the closer the domain $\mathcal{D}$ to the 3D surface $\mathcal{S}$, the lower the area distortion in the parameterization.

### 3.4 CHOICE OF PARAMETRIC DOMAINS

When mapping a 3D surface to a parametric domain, angle and area distortions are usually inevitable. We use the Laplacian regularizer to reduce angle distortion. Given the existence of infinite conformal maps (Jin et al., 2004), identifying those with minimal area distortion is highly desired. Area distortion is influenced by the similarity between the 3D surface and the parametric domain: the closer their alignment, the less the distortion. Thus, a simplistic, closely aligned parametric domain is ideal, balancing simplicity and alignment. Our algorithm accommodates various domains such as spheres, cubes, and polycubes, allowing flexibility in selection. It should be noted that our method supports more complex parametric domains, such as polyhedral complexes Yang et al. (2023), thanks to the representation capability of the SDF sub-network. In our current implementation, we use spheres, cubes and polycubes, mainly because of their simplicity. Figure 3 illustrates the impact on the choice of parametric domain to the parameterization quality.

### 3.5 TRAINING LOSSES

In contrast to NeP (Ma et al., 2022), which necessitate prior information such as tracked mesh and UV mapping for supervision, our method is capable of reconstructing 3D geometry and computing the corresponding texture map directly from multi-view images. This process is supervised by the image loss

$$\mathcal{L}_{\text{rgb}} = \|\mathbf{c}_{\text{gt}} - \mathbf{c}_{\text{pred}}\|_1, \tag{12}$$

where $\mathbf{c}_{gt}$ and $\mathbf{c}_{pred}$ are the ground-truth and predicted colors, respectively. To ensure the shape and appearance latent codes, $\mathbf{z}_s$ and $\mathbf{z}_a$, conform to Gaussian distributions, we incorporate a regularization term, expressed as $\mathcal{L}_{code} = \|\mathbf{z}_s\|_2 + \|\mathbf{z}_a\|_2$ (Park et al., 2019). Additionally, since both the 3D surface $\mathcal{S}$ and the parametric domain $\mathcal{D}$ are represented as signed distance fields, we employ the Eikonal loss (Yariv et al., 2020; Gropp et al., 2020), defined as $\mathcal{L}_{Eik} = (\|\nabla s\|_2 - 1)^2$. Putting it all together, we define the total loss function as follows:

$$\mathcal{L} = \mathcal{L}_{rgb} + \lambda_1 \mathcal{L}_{Eik} + \lambda_2 \mathcal{L}_{cycle} + \lambda_3 \mathcal{L}_{smooth} + \lambda_4 \mathcal{L}_{Lap} + \lambda_5 \mathcal{L}_{shd} + \lambda_6 \mathcal{L}_{code}. \qquad (13)$$

In our implementation, we empirically set the coefficients as $\lambda_1 = \lambda_2 = \lambda_5 = \lambda_6 = 0.01$ and $\lambda_3 = \lambda_4 = 0.001$.

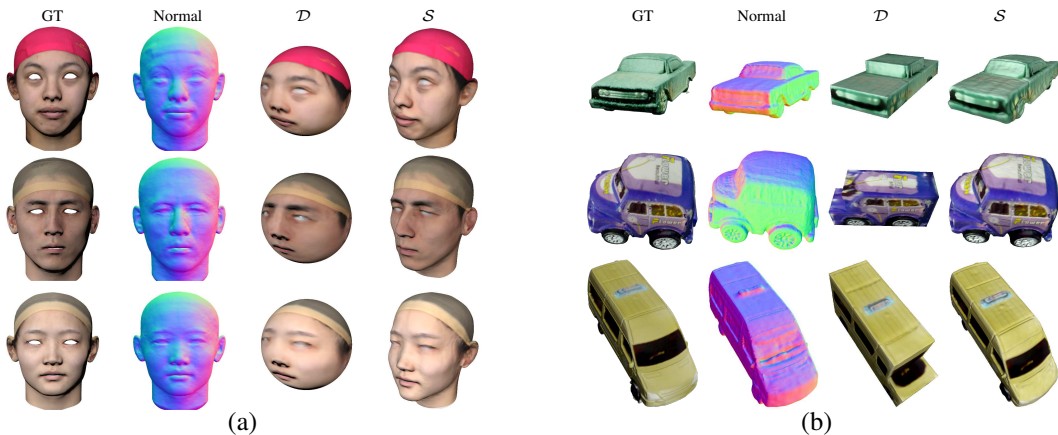

Figure 4: Parameterization results. (a) Human heads are co-parameterized to a sphere owing to the clear geometric resemblance. (b) Given the diverse geometry of cars, each is parameterized to a polycube domain. From left to right: the input image, the normal map of the reconstructed geometry $\mathcal{S}$, the parametric domain $\mathcal{D}$, and the reconstructed surface $\mathcal{S}$. See Appendix A.2 for additional results.

## 4 EXPERIMENTS

**Datasets.** For our experiments, we used a customized dataset named FS-Syn, derived from the Facescape dataset (Yang et al., 2020). From Facescale, we selected 10 human head models and configured 30 fixed viewpoints and lighting conditions for each to obtain synthesized multi-view images. We also evaluated our method on several man-made objects from the DTU dataset (Jensen et al., 2014) and the OmniObject dataset (Wu et al., 2023).

**Training details.** All the sub-networks $F_{def}$, $F_{inv\text{-}def}$, $F_{sdf}$, $F_{mat}$, and $F_{shd}$ are MLPs. See Appendix A.1 for the detailed network architectures. We trained our network using the Adam (Kingma & Ba, 2014) optimizer. Once the parametric domain is chosen, we first trained $F_{sdf}$ for 2,000 iterations to obtain the implicit neural representation of the parametric domain. Subsequently, we fixed $F_{sdf}$ and trained other sub-networks for 2,000 epochs driven by the loss function as Equation (13).

**Results.** Our neural parameterization framework enables the construction of a bijective map between objects and multiple parametric domains such as spheres, cubes, and polycubes, as shown in Figure 4. Additionally, our method supports co-parameterization of multiple objects and facilitates the transfer of material and shading between them. Our neural parameterization algorithm is fully compatible with existing neural rendering pipelines. Given multi-view images as input, we can reconstruct the 3D geometry and create the texture maps simultaneously. As shown in Figure 4, we can obtain high-fidelity results without any prior information.

Compared with NeP (Ma et al., 2022), which is the latest neural surface editing work using neural parameterization, our method eliminates the need for prior information from UV mapping. Unlike our algorithm, NeP learns a map between surfaces and the UV plane. However, it heavily relies on an existing UV mapping as initialization, and yields rather poor texture mapping results without the UV mapping as prior (see Figure 5). Moreover, the results from NeP still exhibit significant distortions

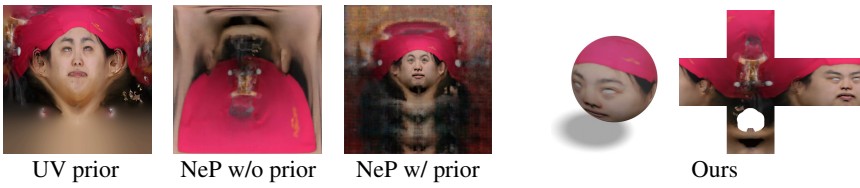

Figure 5: Parametrization results of NeP (Ma et al., 2022)

even with UV prior. Our method can effectively reduce angle distortion by minimizing Laplacian loss and reduce area distortion by choosing an appropriate parametric domain, as demonstrated in Figure 6 (left).

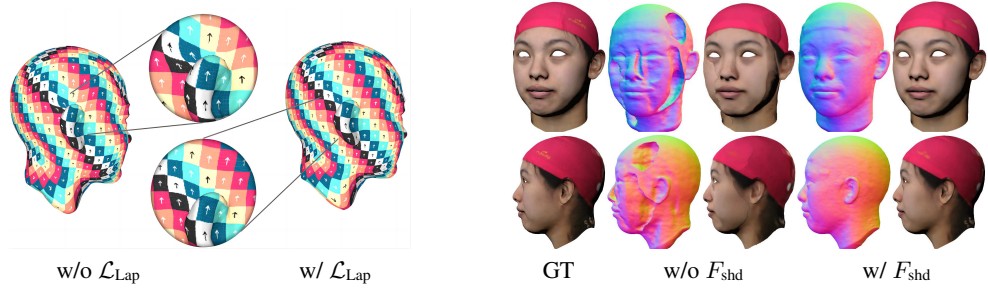

Figure 6: Ablation studies. Left: Utilization of the Laplacian loss has proven effective in reducing angle distortion, especially in regions with high curvature. Right: For scenes with significantly varying illumination, it is crucial to decompose view-dependent shading from the radiance field.

**Radiance decomposition.** In contrast to an entangled radiance field in NeRF (Mildenhall et al., 2020) and VolSDF (Yariv et al., 2021), we decompose the radiance into view-independent material and view-dependent shading according to Equation 9. We illustrate the decomposition results in Figure 7. We would like to emphasize that it is essential to decompose the radiance according to the view-dependent correlations for scenes with varying illumination. As shown in Figure 6 (right), it fails to reconstruct the 3D geometry when we remove the $F_{\text{shd}}$, and the visual quality drops subsequently.

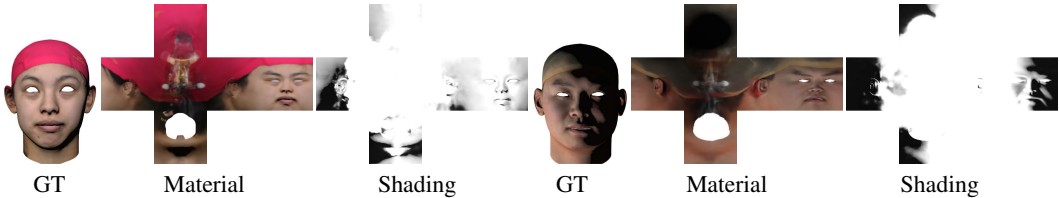

Figure 7: Radiance decomposition results.

**Texture editing.** Both NeuMesh (Yang et al., 2022) and NeP (Ma et al., 2022) can edit the object's texture at a pixel level. We compare the editing and rendering results with NeuMesh and NeP, as shown in Figure 8. NeuMesh requires a mask of the editing region. It also re-trains the network after each edit, which is time-consuming and cannot guarantee the stability of the training results. Our method outperforms NeuMesh in terms of the visual quality. NeP, which is also neural parameterization-based method, requires a UV prior for initialization. It often yields poor parameterization results without such prior. Even with the UV prior, its parameterization results exhibit large distortion, making the edit in the 2D domain challenging. Our method is free of prior, and can yield parameterization with low distortion, thereby facilitating pixel-level editing and material transferring without network re-training.

**Shading editing.** We decompose the 3D radiance field into view-independent material and view-dependent shading defined in the parametric domain. Taking advantage of this decomposition and the embedding into a simple domain, we can edit both the material and shading of the 3D surfaces in an intuitive manner. As illustrated in Figure 9, we modify the shading of one model by transfer-

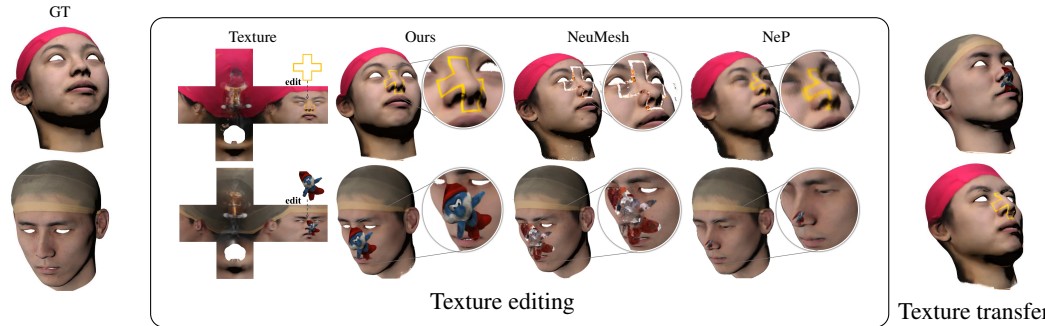

Figure 8: Representing the materials as textures defined in simple parametric domain enables easy editing. Also, our co-parameterization naturally supports texture transfer between different objects.

ring the shading from another model under different lighting conditions, while keeping the material unchanged.

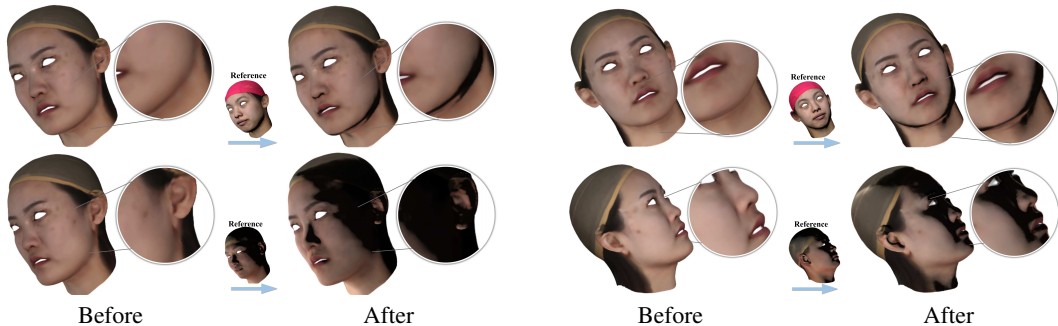

Figure 9: We transfer the decomposed shading from one model to another. See also the accompanying video.

## 5 CONCLUSION

We present a novel neural algorithm for parametrizing 3D surfaces represented by neural implicit functions to a user-defined parametric domain, such as a sphere, cube, or polycube. Utilizing bi-directional deformation, our method is capable of learning a bijective mapping without relying on any prior knowledge, while controlling angle distortion through the usage of a Laplacian regularizer. Furthermore, our approach seamlessly integrates with the neural rendering pipeline, enabling the reconstruction of 3D objects from multi-view images and facilitating efficient volume rendering of modified textures and shadings — eliminating the necessity for network re-training. We demonstrated the efficacy of our method on human heads and man-made objects.

Our method has a few limitations that require further improvement in the future. First, computational results on the FS-Syn dataset show that the reconstruction quality of our method is slightly worse than the standard SDF-based neural rendering algorithms, such as VolSDF (Yariv et al., 2021) and NeuS (Wang et al., 2021). For instance, the PSNRs of our results and VolSDF's are 30.87 and 31.15, respectively. This decline in visual quality arises from the incorporation of additional modules, such as bi-directional deformation and radiance decomposition, resulting in the cessation of updates to the SDF network within our neural parameterization pipeline. Therefore, computing a parameterization without compromising the reconstruction quality is highly desired. Second, our radiance decomposition operates under the assumptions of Lambertian reflectance and grayscale shading, which might hinder its effectiveness when dealing with intricate materials or sophisticated shadings. Third, in our current implementation, we did not optimize the runtime performance. Therefore, training our model is time consuming, requiring 3 days for the 10 human heads of the FS-Syn dataset (Yang et al., 2020). Incorporating recent advancements such as PermutoSDF (Rosu & Behnke, 2023) and Strivec (Gao et al., 2023) has the potential to significantly reduce the training and inference time.

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

## A  APPENDIX

### A.1  IMPLEMENTATION DETAILS

In our framework, the bi-deformation networks $F_{\text{def}}$ and $F_{\text{inv-def}}$, the SDF network $F_{\text{sdf}}$, the texture network $F_{\text{mat}}$, and the appearance network $F_{\text{shd}}$ are all multilayer perceptrons, consisting of 8, 8, 8, 4, and 4 layers, respectively. Each of these hidden layers has 256 neurons. The latent shape codes $z_s$ and the appearance codes $z_a$ are 128-dimensional. Within the neural rendering process, we set the frequency of position encoding for positions and viewpoints as 6 and 4, respectively. In each iteration, we randomly sample 1024 pixels from each input image for training. As for other parameter settings related to neural field reconstruction, we refer mainly to the work of VolSDF (Yariv et al., 2021).

**Baseline implementation details**  For static settings, we remove the warp module in the NeP (Ma et al., 2022). Then we train NeP under two settings, one is under UV ground truth as supervision and the other is not( See Figure 11). The parameter configurations for other settings can be found in the referenced paper.

### A.2 Additional results

Our method is capable of supporting parameterization between more general man-made models and polycubes, as shown in Figure 10(a). We have tested this on the DTU dataset (Jensen et al., 2014). Moreover, our parameterization framework can be used to seamlessly stitch segmented texture maps together, as shown in Figure 10(b). In addition to texture and shading, our framework also can edit geometric details on original models, as illustrated in Figure 10(c).

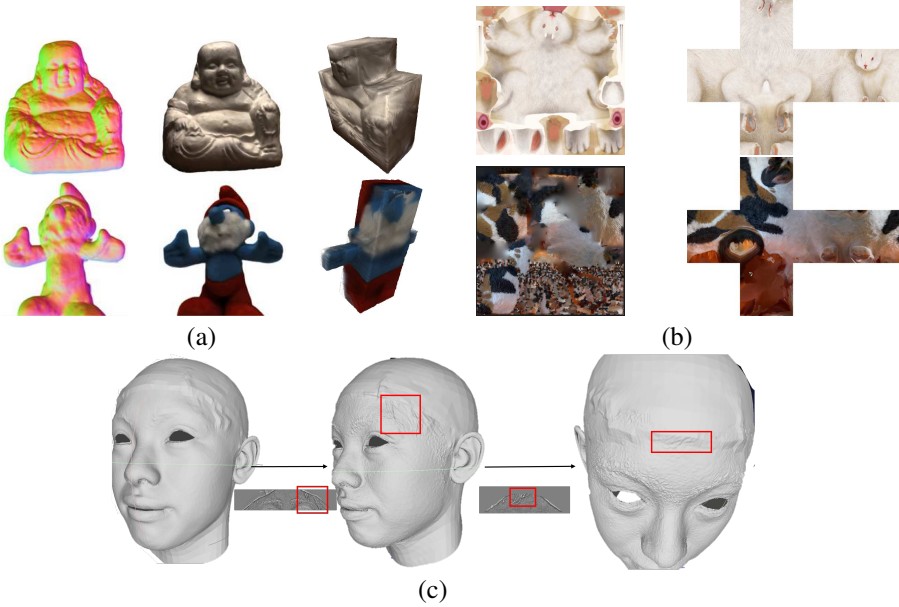

Figure 10: Additional results: (a) normals, rendering and parametric domain on DTU dataset (Jensen et al., 2014) (b) Stitching texture. (c) Geometric detail editing.

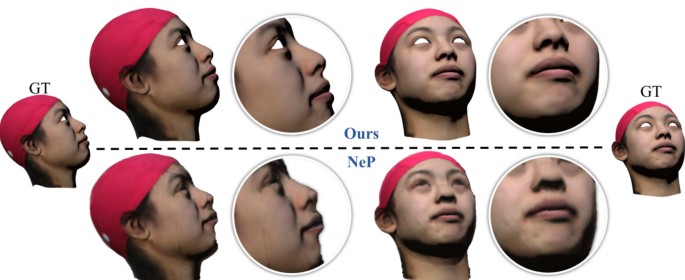

Figure 11: We compare the rendering results with NeP without prior information from UV mapping.

| Method | Repre | Prior | Rendering | Regularizer | Texture | Domain |
|---|---|---|---|---|---|---|
| **NeuTex** (Xiang et al., 2021) | Density | init UV-inv | V.R. | NA | Entangled | Sphere |
| **ISO-UV** (Sagnik Das & Samaras, 2022) | SDF | init UV | D.R. | Jacobian | Entangled | $\mathbb{R}^2$ |
| **NeP** (Ma et al., 2022) | density | Mesh&UV | V.R. | Angle | Disentangled | $\mathbb{R}^2$ |
| **Ours** | SDF | None | V.R. | Laplace | Disentangled | Sphere & polycube |

Table 1: Qualitative comparison with other neural parameterization methods. V.R. and D.R. stand for volume rendering and differentiable rendering, respectively.

### A.3 Qualitative Comparison

We summarize some parametrization methods(refer to Table 1) to illustrate the effectiveness and superiority of our approach. As shown in Table 2, we categorize these methods into three groups:

scene-level, object-level, and pixel-level editing. Scene-level editing methods focus on the entire appearance of a scene including lighting and material. Object-level editing methods employ distinct part-based latent codes to manipulate different attributes of a scene, such as hairstyles in i3DMM (Yenamandra et al., 2021). Pixel-level editing, on the other hand, offers a fine-grained editing result by taking into account precise user guidance.

| Method | Level | Retraining | 3D input |
|---|---|---|---|
| **EditNeRF** (Liu et al., 2021) | Object | Y | N |
| **StylizedNeRF** (Huang et al., 2022) | Scene | N | N |
| **i3DMM** (Yenamandra et al., 2021) | Object | N | Y |
| **NeuMesh** (Yang et al., 2022) | Pixel | Y | N |
| **Seal3D** (Wang et al., 2023) | Pixel | Y | N |
| **NeuTex** (Xiang et al., 2021) | Pixel | N | Y |
| **NeP** (Ma et al., 2022) | Pixel | N | Y |
| **Ours** | Pixel+Object | N | N |

Table 2: Qualitative comparison with other neural texture editing methods.

