# OpenReview forum: "Bi-directional Deformation for Parameterization of Neural Implicit Surfaces"
_ICLR.cc/2024/Conference — ICLR 2024 Conference Withdrawn Submission_

### Official Review · Reviewer_4YC2 · 2023-10-31

**Soundness:** 3 good
**Presentation:** 2 fair
**Contribution:** 1 poor
**Rating:** 3
**Confidence:** 4

**Summary:**

This paper proposes an algorithm for implicit surface editing. Typically, relying on the NeRF pipeline, this paper introduce the neural surface parameterization that makes editing much feasible in 2D domain. To do so, this paper focuses on the bi-directional deformation design which maps the 3D space and 2D map.

**Strengths:**

It is interesting to apply surface parameterization into the neural implicit surface. Such trick is widely used when modeling 3D human from 2D UV texture map. So the proposed direction is reasonable and meaningful approach.

To buildup this model, this paper designs bi-directional deformation modules. Similar to the space deformation proposed by D-NeRF (CVPR 2021), Nerfies: Deformable Neural Radiance Fields (CVPR 2021), DeVRF: Fast Deformable Voxel Radiance Fields for Dynamic Scenes (Neurips 2022), and Spacetime Surface Regularization for Neural Dynamic Scene Reconstruction (ICCV 2023).

**Weaknesses:**

Despite the novel idea, I am not satisfied with several issues.

W-1. Poor writing.
First of all, there are huge mismatch between the title, the abstract, and the content of this paper. In the title and the abstract section, the neural editing task is treated as a minor subtask. Even, in the title, there are no words related to the task itself. However, when I started to read the introduction section, this paper fully focused on the neural editing task. Such a contradiction is confusing me a lot.

W-2. Confusing contribution.
Similar to the problem in W-1, I cannot clearly catch the contribution of this paper. What is the main contribution of this paper? Neural surface parametrization or bi-directional deformation? I am pretty sure that when people initially see the title and start to read down the introduction and methodology section, they will get lost.

The title indicates that the contribution is mainly from the bi-directional deformation design. However, as I stated in the Strength slot, there are so many papers that utilized the bi-directional deformation in the NeRF framework. Moreover, __I cannot clearly tell the difference between the elastic loss in the Nerfies paper (Nerfies: Deformable Neural Radiance Fields, CVPR 2021) and the proposed Laplacian regularization.__

If the neural surface parameterization is the dominant contribution, the organization of the paper is really bad. I am really not satisfied with the writing itself.

W-3. Lack of experiements.
As the authors stated in the related work section, there are some previous works that handle the neural editing task. So the clear difference and experiments are truly needed to verify the efficacy of the proposed contributions. However, the experiments that are mainly used by the authors are quite limited. __I want the authors to provide tons of qualitative/quantitative comparison with recent studies__

Moreover, is there any ablation study for the neural surface parametrization? __The lack of ablation study limits the reviewers to judging the effectiveness of the proposed modules__

**Questions:**

Please refer to the Weakness slot. Despite the novel idea, experiments are not that good enough to support the concreteness of this paper. Moreover, the quality of writing is disappointing me a lot.

**Details Of Ethics Concerns:**

No ethics review.

---

### Official Review · Reviewer_3Lvc · 2023-10-31

**Soundness:** 2 fair
**Presentation:** 3 good
**Contribution:** 2 fair
**Rating:** 5
**Confidence:** 3

**Summary:**

This paper addresses the topic of parameterization of neural implicit surfaces. The core method in this paper is a bi-directional mapping, achieved by learning two (a forward and a backward) deformation fields (mappings) represented by MLPs, between the parametric domain (usually a simple geometric primitive such as a sphere or a cube) and the target object shape domain. A cycle-consistency loss is applied to encourage the two mappings to be bijective. A Laplacian loss is introduced to minimize distorsion after mapping to the parametric domain. Since the 3D shape can be parameterized to a 2D surface, the authors further propose to decompose appearance into view-independent albedo and view-dependent shading.
Experiments have validated the designed characteristics of the method: flexible and effective parameterization of the source shape, reduction of angle distorsion, decomposition of appearance, and the capability of editing texture/shape, although no quantitative evaluations are shown.

**Strengths:**

- The method is intuitive and very clearly demonstrated. Using a cycle-consistency to regularize the forward and backward mappings is a good idea.
- From the qualitative results, the proposed method indeed has the capability of decomposing texture and shading, and the shape editing has a higher robustness and less visual artifacts than the state-of-the-art (NeuMesh and NeuP). Considering that the method does not rely on manually-defined UV maps, these characteristics have a great potential for practical applications.
- The paper is well-written and the method description is very clear. Especially, I appreciate the well-written related work section that provides a good review of the literature tracing back to early 2000s.

**Weaknesses:**

- My biggest concern is the lack of quantitative evaluation, and hence my rating for the 'soundness' part at this iteration. I understand that the focus of this paper is on the texture editability enabled by the proposed method. However, quantitative analysis on other aspects can still better help the reader understand the capability of the method. For example, the geometry reconstruction errors could have been reported after applying the forward and backward mappings (this evaluates how well the cycle loss works and the smoothness of the learned deformations). For the appearance decomposition, the albedo (texture) PSNR, and the re-relighting PSNR are commonly used in the inverse rendering literature and would have been interesting to see. I'm not very much familiar with the metrics for distortion and cannot comment on that aspect (sorry for that). For texture editing/swapping/painting experiments I agree that mostly qualitative demonstrations are commonly seen in literature and it is fine (though I encourage the authors display more examples).
- From the description of the paper, the source parametric domain need to have the same genus as the target shape, but for some complicated geometry, knowing the genus isn't trivial. Also, the choice of the source parametric domain in terms of shape geometry and topology is subject to manual heuristics (authors please correct me if wrong).
- Only relatively simple geometry are shown in the applications. For conversions between sphere-to-head and cube-to-van, not much amplitude of the deformation is required to map from the source to target shape. How does the proposed method perform on more complicated cases? A qualitative example on such more challenging cases (for example the Dragon from Stanford 3D Scanning Repository) can reveal the capability and limitations of the learned mappings and how they compare with more traditional UV maps (which involve more manual efforts but can be tailored to be precise).
- The loss in Eq. (8) will encourage the tangential component in the Laplacian to vanish. Why would that encourage preservation of angles? A bit more elaboration at the end of Sec. 3.2 would be appreciated.

**Questions:**

- Since the proposed method can be flexible with the choice of the source parametric domain (e.g. sphere/cubes both work for cars, Fig.3),  does it still ensure the correspondence across different views or timesteps (for sequential data)? For example, if one chooses a sphere to parameterize a dynamic head geometry captured in a sequence with multi-view setup. Would there be texture inconsistency at different views? Would there be texture sliding/flickering across different frames?
- Would the proposed method generalize to larger scale scenes, e.g. room scans?
- For more complex geometry, is there a general guideline for choosing the source shape? For example, what would be a good parametric domain for the Stanford Lucy model?

---

### Official Review · Reviewer_bgLy · 2023-11-01

**Soundness:** 1 poor
**Presentation:** 1 poor
**Contribution:** 1 poor
**Rating:** 3
**Confidence:** 3

**Summary:**

This paper presents a method for parametering neural 3D implicit surfaces into a 2D parametric domain, including a sphere, cube, or polycube. The goal is to establish a bidirectional mapping between 3D surface and the 2D domain. This work claims to achieve the bijective mapping without relying on any prior knowledge and proposes to reduce angle distortion using a Laplacian regularizer. It also claims to support seamless integration with neural rendering pipelines and can provide intrinsic radiance decomposition which generates view-independent material editing and view-dependent shading editing.

**Strengths:**

- The 2D parameterization results look reasonable.
- The texture transfer results generated by the proposed method look more natural and achieve better transition smothness than the other candidate approaches.

**Weaknesses:**

- The writing of the method part is very difficult to follow - I think there are lots of technical details missing or not accurately presented. I will detail below and in the "Questions" section.

- The proposed method is not very technically sound. There are many flaws in the presented solutions - here I just name the major ones:
1) the 3D surface point p and its projection p' at the 2D parametric domain do not have the same dimension. How could p' computed from p by simply adding a displacement vector as shown in Equation (1)?
2) Though an L2 smoothness loss is proposed to regularize the norm of the displacement vector, there is no explicit/physical term to ensure the content in the 2D domain is continuous as in 3D. For instance, a 3D face remains a 2D face (might be distorted) in the parametric domain.
3) For the appearance decomposition, the method only proposes a local shading constraints to minimize the L1 norm of the shading component. I don't understand how this simple term can achieve a good decomposition when there is not restriction on the material side.

- The method claims to work on polycube domain. However, there are no such results to support this claim.

- The results are only limited to a couple of faces and very simple objects like cars. It would be better to verify its performance on more complex objects.

**Questions:**

- Why the method cannot support a regular 2D square parametric domain which is the most commonly used domain in parameterization?

- What does it mean that your framework involves two SDFs? I understand one is the 3D surface, what about the other one of the parametric domain?

- The input of the method are only multiview images. How could you determine the surface in the beginning? I didn't see any part of the paper mentions training of NeRF/NeuS. Is anything missing?

---

### Official Review · Reviewer_sy5Q · 2023-11-08

**Soundness:** 3 good
**Presentation:** 2 fair
**Contribution:** 3 good
**Rating:** 5
**Confidence:** 4

**Summary:**

This paper proposes an efficient representation for implicit surfaces. Specifically, the proposed method converts implicit surfaces with radiance fields (e.g., NeuS) into deformation from pre-defined primitive shapes. This enables baking the material and reflectance (as well as deformation, perhaps) information onto 2D images. The proposed method also introduces Laplacian loss as a regularizer to mitigate the distortion. Experiments show that the 2D-based representation achieves plausible and easy editing on reconstructed surfaces.

**Strengths:**

+ The representation would technically sound. Baking detailed information onto simple proxy shapes is a traditional yet effective way, and this work correctly brings it to neural implicit surface representation.
+ Editing results are, qualitatively, better than SOTA editing methods of neural fields.

**Weaknesses:**

- A straightforward approach of material editing is, specifically, if they use implicit surface representation to convert the reconstructed 3D representation to meshes and then edit UV textures (or whatever). Compared with those \emph{standard} approaches, the motivation for directly editing NeuS-like representation is not quite clear to me.

- An important drawback of the proposed method is the degradation of the reconstructed shape compared with usual neural implicit surfaces, which can be clearly seen in, e.g., Fig4(b).

- Related to the above, this paper does not have any quantitative evaluations. At least, readers should wonder about the effect of shape degradation quantitatively. Since they use the DTU dataset, they could do quantitative evaluations on shape reconstruction. For example, the most recent paper proposing NeP (Ma et al. 2022) shows that their method achieves a mostly comparable (while slightly worse) novel-view synthesis accuracy to NeRF-like methods.

- The paper only tested their method for faces and cars (and a few objects in supplementary materials). There should be well-known object categories with more diverse shapes, such as chairs, sometimes involving topology changes. To show the generalizability of the proposed method, many readers may want to see such challenging targets.

- Shape editing is another important scenario that is not mentioned in the paper (and NeP also actually handles the shape editing tasks). I consider this to be possible since their representation includes the displacement maps.

**Questions:**

Q. Is this method not quite good at shape editing purposes?